# Relevance of A Disintegrin and Metalloproteinase Domain-Containing (ADAM)9 Protein Expression to Bladder Cancer Malignancy

**DOI:** 10.3390/biom12060791

**Published:** 2022-06-06

**Authors:** Michika Moriwaki, Trang Thi-Huynh Le, Shian-Ying Sung, Yura Jotatsu, Youngmin Yang, Yuto Hirata, Aya Ishii, Yi-Te Chiang, Kuan-Chou Chen, Katsumi Shigemura, Masato Fujisawa

**Affiliations:** 1Department of International Health, Kobe University Graduate School of Health Sciences, 7-10-2 Tomogaoka, Suma-ku, Kobe 654-0142, Japan; m.ichigan3012g@gmail.com (M.M.); yura199911.kcao@gmail.com (Y.J.); tyanhira0430@gmail.com (Y.H.); ayacchibonbon@gmail.com (A.I.); 2International Ph.D. Program in Medicine, College of Medicine, Taipei Medical University, Taipei 11031, Taiwan; lehuynhtrang2010@gmail.com; 3International Ph.D. Program for Translational Science, College of Medical Science and Technology, Taipei Medical University, 250 Wu-Hsing st, Taipei 11031 Taiwan; ssung@tmu.edu.tw; 4Department of Urology, Kobe University Graduate School of Medicine, 7-5-1 Kusunoki-cho, Chuo-ku, Kobe 650-0017, Japan; yym1112@gmail.com (Y.Y.); masato@med.kobe-u.ac.jp (M.F.); 5Department of Urology, Taipei Medical University Shuang Ho Hospital, 291 Zhongzheng Road, Taipei 23561, Taiwan; emilchiang@gmail.com (Y.-T.C.); kuanchou@tmu.edu.tw (K.-C.C.)

**Keywords:** A Disintegrin and Metalloproteinase Domain-Containing (ADAM)9 protein, bladder cancer, epithelial–mesenchymal transition, migration, malignancy

## Abstract

We evaluated the effect of A Disintegrin and Metalloproteinase Domain-Containing (ADAM)9 protein on exacerbation in bladder cancer KK47 and T24. First, we knocked down ADAM9 and investigated cell proliferation, migration, cell cycle, and the epithelial–mesenchymal transition (EMT)-related proteins expression in vitro. We then investigated the expression level of ADAM9 in clinical urine cytology samples and the Cancer Genome Atlas (TCGA) data. Cell proliferation was significantly reduced in both cell lines after ADAM9 knockdown. In the cell-cycle assay, the percentage of G0/G1 cells was significantly increased in ADAM9 knockdown T24. Migration of T24 was more strongly suppressed than KK47. The expression level of EMT-related proteins suggested that EMT was suppressed in ADAM9 knockdown T24. TCGA analysis revealed that ADAM9 mRNA expression was significantly higher in stage IV and high-grade cancer than in other stages and low-grade cancer. Moreover, in the gene expression omnibus (GEO) study, bladder cancer with surrounding carcinoma and invasive carcinoma showed significantly high ADAM9 mRNA expression. We found that ADAM9 knockdown suppressed cell proliferation and migration in bladder cancer and that high-grade bladder cancer is correlated with higher expression of ADAM9.

## 1. Introduction

Bladder cancer is classified into non-muscle-invasive and muscle-invasive types. Non-muscle-invasive cancer is treated by transurethral resection, intravesical chemotherapy, and/or immunotherapy. Muscle invasive cancer is typically treated with radical cystectomy because of higher rates of progression and recurrence [1]. Muscle-invasive bladder cancer has a worse prognosis and lower quality of life (QOL).

Non-muscle-invasive cancer is largely composed of malignant epithelial cells, while muscle-invasive cancer cells are more mesenchymal. Epithelial-to-mesenchymal transition (EMT) drives the progression of epithelial cells to a more mesenchymal configuration. In EMT, cancer cells are released from tight adhesion, lose polarity, and undergo changes in the cytoskeletal structure, allowing the cells to transform into an invasive phenotype. Cancer cell EMT is regulated by physical stresses internal to the tumor, such as hypoxia, and by stimulation external to the tumor, such as inflammatory cytokines [2]. Cancer cells that have undergone EMT have molecular characteristics indicative of high metastatic potential. High metastatic ability tends to emerge in the later stages of bladder cancer, compared with other cancer types. This suggests that bladder cancer cells acquire their high metastatic ability through EMT [3]. Since Vimentin and N-cadherin are increased in cells progressing to EMT, while E-cadherin protein decreases, it can be considered that EMT is suppressed when Vimentin and N-cadherin are suppressed, whereas E-cadherin is not. Suppressing EMT may help prevent bladder cancer progression since several previous studies have suggested that EMT is associated with bladder cancer progression and metastasis [4,5].

A Disintegrin and Metalloproteinase Domain-Containing (ADAM)9 protein is drawing attention as a new therapeutic target in cancer. Disintegrin is a generic term for platelet aggregation inhibitory factors that bind to integrins such as αVβ3 and α5β1 present in cell membranes, for instance, melanoma cells and vascular endothelial cells [6]. These cells are competitively blocked from binding to an attachment factor, so disintegrins have the potential as powerful inhibitors of cancer cell migration and neovascularization [7]. Metalloproteases with disintegrin as a domain structure have been found in various biological species ranging from mammals (including humans) to nematodes, and Drosophilidae, and have come to be referred to as the ADAM family [8], of which ADAM9 is a member.

ADAM9 has been studied in several cancers. In prostate cancer, ADAM9 cleaves insulin β-chain, tumor necrosis factor (TNF)-α, transforming growth factor (TGF)-α, gelatin, β-casein, etc., and induces the shedding of epidermal growth factor (EGF), fibroblast growth factor receptor 2 (FGFR2)-IIIB and heparin-binding EGF-like growth factor. Therefore, ADAM9 causes the degradation of specific substrates, releases active growth factors, and interacts with key regulatory factors, and its expression is related to several cancer processes, including cell growth and invasion [9,10]. However, it has been less widely studied in bladder cancer. The purpose of this study was to investigate the involvement of ADAM9 in the exacerbation of bladder cancer and to clarify whether ADAM9 can be a new therapeutic target for preventing bladder cancer cell EMT.

## 2. Materials and Methods

### 2.1. Cells

We used human urinary bladder cancer cell lines KK47 and T24. KK47 was generously provided by Dr. Naito (Kyusyu University, Fukuoka, Japan), and T24 was purchased from the American Type Culture Collection (Manassas, VA, USA). KK47 is derived from non-muscle-invasive bladder cancer, and T24 is derived from muscle-invasive bladder cancer.

### 2.2. Cell Culture

Both cell lines were cultured in Roswell Park Memorial Institute (RPMI) 1640 medium (Thermo Fisher Scientific, Waltham, MA, USA) complemented with 10% fetal bovine serum, at 37 °C with 5% carbon dioxide.

### 2.3. ADAM9 Knockdown

ADAM9 was knocked down by the ON-TARGET plus SMARTpool siRNA against ADAM9 (Thermo Fisher Scientific, Waltham, MA, USA) with Lipofectamine 2000 (Invitrogen, Carlsbad, CA, USA) [11]. We used a final siRNA concentration of 5 nM. The expression of ADAM9 was confirmed by RT-qPCR. RNA was extracted from cells after 72 h of culture with the addition of reagents. Total RNA was extracted according to the manufacturer’s instructions using the NucleoSpin RNA kit (Takara Bio, Kusatsu, Japan). Complementary DNA was synthesized using a ReverTra Ace qPCR RT kit (TOYOBO, Osaka, Japan). Primer sequences are shown in Table 1. For the control group, cells that had not been subjected to ADAM9 knockdown treatment were used.

### 2.4. Cell Proliferation Assays

The effects of ADAM9 on cell proliferation in KK47 and T24 cells were measured using Cell Titer 96 AQueous One Solution Cell Proliferation Assay (MTS) (Promega, Madison, WI, USA). After ADAM9 knockdown, we seeded cells on 96-well plates at 2 × 10^3^ cells/well (n = 3), and cultured them for 24 h, 48 h, and 72 h, and the absorbance was measured at 490 nm [12].

### 2.5. Cell-Cycle Assays

The effects of ADAM9 on the cell cycle in KK47 and T24 cells were measured using Cell Cycle Assay Solution Blue (DOJINDO, Kamimashiki, Japan). After ADAM9 knockdown, 1mL cell suspension having 1 × 10^5^ cells/mL was prepared (n = 3) and centrifuged at 300 g for 5 min. The supernatant was removed, 1 mL PBS was added, and after resuspension, it was centrifuged at 300 g for 5 min. The supernatant was removed again; then, 0.5 mL PBS and 5 μL cell-cycle assay solution were added. After incubation for 15 min at 37 °C in shade, cells in each cell cycle were measured via flow cytometry.

### 2.6. Wound-Healing Assays

Cell migration was analyzed via wound-healing assay. We scraped confluent cells (n = 3) with a micropipette tip and acquired images at 0 h (control), 6 h, 12 h, and 18 h using the EOS utility (Canon, Ota, Japan). The wound area was measured using ImageJ (Java, Bethesda, MD, USA), and we compared the total wound area after 6 h, 12 h, and 18 h under control and ADAM9 knockdown conditions [13].

### 2.7. RT-qPCR

We used cDNA as the template for RT-qPCR with a PCR kit (Takara Biotechnology Co., Ltd. Kusatsu, Japan), with β-actin as an internal control (n = 3). Primer sequences are shown in Table 1. We pursued the relationship between EMT and ADAM9 in bladder cancer cells by comparing the gene expression of EMT-related proteins between knockdown and control groups. We targeted Vimentin, N-cadherin, and E-cadherin. Primer sequences are shown in Table 1. The PCR conditions were as follows: 94 °C for 2 min followed by 30 cycles of 94 °C for 1 min, 61 °C for 1 min, and 74 °C for 1 min and 30 sec.

The relative expression levels of RNA were analyzed using *2*^–ΔΔCt^ values [12]. To correct for differences in the amount of DNA added for each sample and to reduce variation caused by PCR setup and the cycling process, β-actin, which is a housekeeping gene and its expression levels remain relatively stable in response to any treatment, was used as a reference gene [13]. With respect to the ΔΔCt of the 2- ΔΔCt method, the first ΔCt is the difference in the threshold cycle between the target and reference genes (1).
ΔCt = Ct (a target gene) − Ct (a reference gene).(1)

Next, calculate the difference between ΔCt of the reference sample and ΔCT of other samples to be compared (2).
ΔΔCt = ΔCt (a target sample) − ΔCt (a reference sample) (2)

The final result of this method is presented as the fold change in target gene expression in a target sample relative to a reference sample, normalized to a reference gene. The relative gene expression is usually set to 1 for reference samples because ΔΔCt is equal to 0 and therefore 2^0^ is equal to 1.

### 2.8. TCGA Analysis

The gene expression profiles of mRNA and related clinical information for bladder cancer patients were downloaded from cBioPortal accessed on 6 January 2022. (http://www.cbioportal.org/). The Bladder Urothelial Carcinoma (TCGA (n = 408), Firehose Legacy 9) was used to compare the ADAM9 expression with stage, grade as well as ADAM10 and ADAM17.

### 2.9. GEO Study

RNA-seq data available in the Gene Expression Omnibus were utilized to evaluate the mRNA levels of ADAM9 (GSE3167 (dataset record GSD1479)). We collected bladder cancer clinical information (n = 60) in terms of surrounding carcinoma or not, and invasive carcinoma or not. We analyzed the data to investigate the relationship between the migration of bladder cancer cells and the expression of ADAM9.

### 2.10. Statistical Analysis

We used Student’s *t*-tests to compare controls to targets in the proliferation assay, cell-cycle assay, wound-healing assay, and RT-qPCR. The same statistical method was also used in the TCGA analysis to compare cancer to no cancer, and high-grade cancer to low-grade cancer. We used EZR (Jichi Medical University Saitama Medical Center, Saitama, Japan) for statistical analysis. In the TCGA analysis and GEO study, ADAM9 mRNA expression at each stage of bladder cancer and in each cancer type was evaluated with one-way analysis of variance (ANOVA) using Prism 8.4.0 software (GraphPad, CA, USA). For all tests, statistical significance was defined as *p* < 0.05.

In summary of the Methods section, we knocked down ADAM9, after which the effects were observed on (1) cell proliferation, (2) cell cycle in order to see which cell-cycle step in ADAM9 plays a role, (3) wound-healing assay in order to see how ADAM9 affects cell invasion or EMT, (4) EMT marker expressions via RT-PCR (4 kinds of analysis). Next, using databases such as TCGA analysis and GEO study in bladder cancer cell lines, we examined how ADAM9 mRNA is expressed in higher-stage and higher-grade bladder cancer samples.

## 3. Results

### 3.1. ADAM9 Knockdown

ADAM9 expression status was confirmed with RT-PCR at 72 h in only control cancer cell lines, not ADAM9 knockdown cancer cell lines. ADAM9 was significantly reduced in ADAM9 knockdown cells, compared with control cells (*p* < 0.001) (Figure 1).

### 3.2. Proliferation Assay

Cell proliferation was significantly reduced by ADAM9 knockdown, compared with control treatment at 48 h (*p* < 0.001) and 72 h (*p* < 0.001) in KK47 cultures. At 72 h, proliferation decreased by 34% compared with controls. Cell proliferation was significantly reduced by ADAM9 knockdown compared with control treatment at 48 h (*p* = 0.003) and 72 h (*p* < 0.001) in T24. At 72 h, proliferation decreased by 27% compared with control cells. This suggested an association of knockdown of ADAM9 with inhibition of bladder cancer growth (Figure 2).

### 3.3. Cell-Cycle Assay

KK47 cells showed no significant difference between control and ADAM9 knockdown in each phase. In T24, the percentage of cells in the G0/G1 phase was significantly increased in ADAM9 knockdown cells, compared with control cells (*p* < 0.001), and cells in the S phase significantly decreased with ADAM9 knockdown, compared with control (*p* < 0.001) (Figure 3). This suggested that ADAM9 may accelerate the cell cycle by promoting DNA synthesis.

### 3.4. Wound-Healing Assay

At 18 h of culture, there was a significant difference in wound area between control KK47 cells (72.8 ± 1.78%) and ADAM9 knockdown cells (89.4 ± 12.5%) (*p* = 0.021). ADAM9 knockdown also inhibited wound closure at 6 h (*p* = 0.024), 12 h (*p* = 0.020), and 18 h (*p* = 0.004) in T24 cells. At 18 h there was a significant difference in wound area between control cells (54.3 ± 4.11%) and ADAM9 knockdown cells (82.9 ± 16.6%) (Figure 4), suggesting that ADAM9 may be associated with bladder-cancer cell migration.

### 3.5. RT-qPCR

Vimentin was significantly reduced in ADAM9 knockdown cells, compared with control cells (*p* < 0.001) in the KK47 cell line. N-cadherin and E-cadherin were elevated in ADAM9 knockdown cells, compared with control cells (N-cadherin: *p* = 0.016, E-cadherin: *p* = 0.003). Vimentin and N-cadherin were reduced in ADAM9 knockdown cells, compared with the group of control cells (Vimentin: *p* = 0.011, N-cadherin: *p* = 0.0011) in T24 cells. E-cadherin was elevated in ADAM9 knockdown cells, compared with control cells (*p* < 0.001) in T24 cells. These results strongly suggested that ADAM9 knockdown suppressed EMT in both KK47 and T24 cells (Figure 5, Table 2).

### 3.6. TCGA Analysis

The average level of ADAM9 mRNA in stage IV bladder cancer patients was significantly higher than patients in stages I, II, and III (3094.24 vs. 2083.05, 2302, 2403.38, respectively) (*p* = 0.014) (Figure 6A). ADAM9 mRNA expression in high-grade bladder cancer was significantly higher than in low grade (*p* = 0.031) (Figure 6B). The analysis of mRNA levels from the TCGA database also realized that the higher expression of ADAM9 and ADAM10 was significant, compared with ADAM17 (Figure 6C). ADAM10 mRNA expression was high in stage IV but had no difference in low and high grades of histology (Figure 6D,E).

### 3.7. GEO Study

We classified the data as to surrounding carcinoma and invasive carcinoma into five groups. ADAM9 expression was highly and significantly expressed in bladder cancer with surrounding carcinoma or invasive carcinoma (*p* < 0.001) (Figure 7), suggesting that ADAM9 may be associated with the promotion of cell migration.

## 4. Discussion

Proliferation and migration significantly decreased with ADAM9 knockdown, compared with control, and EMT was also suppressed in both cell lines in this study of the effects of ADAM9 knockdown. Results suggested a possible correlation between exacerbation of bladder cancer and the expression of ADAM9. For suppression of proliferation, there were no clear differences between KK47 and T24, but KK47 cell migration was significantly reduced only after 18 h, compared with T24 after 6 h. In terms of EMT and ADAM9, N-cadherin in KK47 knockdown cells increased compared with control, which did not suggest suppression of EMT, while N-cadherin in T24 decreased with ADAM9 knockdown compared with control, suggesting suppression of EMT. Since T24 is muscle-invasive bladder cancer, the results suggested that invasive bladder cancer cells are more dependent on ADAM9 signaling for progression.

Several previous studies have investigated the relationship between ADAM9 and cancer proliferation. These studies suggested that proliferation was suppressed by ADAM9 knockdown in lung cancer, pancreatic ductal adenocarcinoma, and gastric cancer [9,10,14]. These studies often showed significant differences at 96 h of ADAM9 knockdown, whereas we found a significant difference at 48 h in this study. This suggested that the inhibitory effect of ADAM9 knockdown on bladder cancer cell proliferation may be relatively high, compared with other cancer types.

Cell-cycle regulation is one of the important mechanisms of antiproliferation in cancer [15]. In this study, ADAM9 knockdown significantly induced cell-cycle accumulation at G0/G1 phase in T24, and similar results have been observed in other types of cancer [16]. The G0 phase is the period during which cell proliferation is dormant, and the G1 phase is the pre-stage of cell proliferation, so it is suggested that cell proliferation was suppressed by increasing the number of cells during the period when cell proliferation was not performed by ADAM9 knockdown [16].

A previous study conducted in relation to pancreatic cancer, where EMT is considered a prognostic factor as it is in bladder cancer, suggested that ADAM9 knockdown attenuated cellular migration in Panc-1 and ASPC-1 cells [17]. Suppression of migration by ADAM9 knockdown has been reported in the PC3 prostate cancer cell line used to study cell adhesion, spreading, and migration, and in breast cancer [18,19,20]. Many other studies have reported that ADAM9 knockdown suppressed migration in cancer cells compared with normal cells [18,19,20]. This study is in accord with other cancer studies but also reports a new finding that the effect of ADAM9 knockdown was more pronounced in muscle-invasive cancer than in non-muscle-invasive cancer.

The proteins related to EMT (Vimentin, N-cadherin, E-cadherin) have been investigated in several studies. Vimentin and N-cadherin are markers of mesenchymal cell lines, and E-cadherin is a marker of epithelial cell lines. At the molecular level, EMT is characterized by upregulation of Vimentin and N-cadherin and downregulation of E-cadherin [21,22]. In pancreatic ductal adenocarcinoma, ADAM9 knockdown cells had increased expression levels of E-cadherin compared with controls [23]. In prostate cancer, Vimentin, N-cadherin, and E-cadherin fluctuated in a manner suggesting the suppression of EMT via regulation of ADAM9 [24]. In this study, only N-cadherin levels in KK47 did not suggest suppression of EMT. A lung cancer study reported that the correlation between N-cadherin and ADAM9 was stronger in highly malignant cancers [25]. Our results here also seem to show a more pronounced effect of ADAM9 knockdown in highly malignant T24.

ADAM proteins have been reported to play important roles in the proteolytic release of transmembrane proteins, including the ectodomain of major histocompatibility complex class I-related chain A (MICA) [26]. ADAM9 knockdown experiments have revealed that ADAM9 is involved in MICA ectodomain shedding of human hepatocellular carcinoma (HCC) cells before [27]. MICA, which is shed from tumor cells, plays a critical role in the immune surveillance against tumor cells and is associated with the prognosis of several malignancies [28]. Therefore, it is suggested that shedding of MICA by ADAM9 may be one of the mechanisms in cell proliferation inhibitory effect and EMT inhibitory effect and the relationship between malignancy and ADAM9 expression obtained in this experiment.

Our findings suggest that ADAM9 may be a new therapeutic target as well as a marker of poor prognosis in bladder cancer. If ADAM9 knockdown suppresses progression from non-muscle-invasive to muscle-invasive bladder cancer, we could reduce the need for radical cystectomy and improve patient QOL. There is also a possibility that the mortality rate would decrease if metastasis from muscle-invasive bladder cancer to multiple organs could be suppressed. Other studies have also found that ADAM9 and microRNA-126, which have been reported to upregulate ADAM9, may be new therapeutic targets and markers for cancer [24,29].

Based on the results of TCGA analysis, the higher the stage of the bladder cancer, and the more malignant it is, the more ADAM9 mRNA is expressed. In addition to ADAM9, ADAM10 and ADAM17 are also known to predict poor prognosis in hepatocellular carcinoma and lung cancer, based on previous studies [30]. Therefore, in this study, we also analyzed the association between these mRNA expression levels and bladder cancer patients. However, ADAM17 showed no significant increase in expression. Since ADAM10 showed increased mRNA expression, we analyzed ADAM10 mRNA levels by stage and grade and found no association between these and expression levels. Therefore, ADAM9 may be a useful prognostic marker, especially in that it correlates with bladder cancer stage classification and grade. Moreover, our GEO study found that the group with superficial transitional cell carcinoma with surrounding carcinoma in situ lesions, or the invasive carcinoma group had higher expression of ADAM9, further supporting the cell migration promoting the function of ADAM9. Superficial transitional cell carcinoma accounts for a large portion of all bladder cancer. Although this is less malignant than invasive cancer, cancer cell migration can occur, especially when the lesion is surrounded by carcinoma in situ.

This study has several limitations. First, it does not include a mechanistic study related to EMT or migration. Second, additional bladder cancer cell lines might be informative. Third, we did not perform an animal study. Fourth, our story of ADAM9 knockdown and EMT may be in part conclusive because Yang et al. stated that EMT status cannot be assessed on the basis of one or a small number of molecular markers [31]. These limitations need to be addressed in future studies.

Many studies have reported an association between ADAM9 and cancer, including bladder cancer. Although ADAM9 has been proposed as a therapeutic target, there have been few reports on specific potential treatments. In the future, we will investigate potential treatments in vivo using a mouse model. In this study, we proved that there is a significant difference in ADAM9 expression level depending on the grade of cancer. However, it is necessary to determine the cut-off value in terms of specificity and sensitivity to use ADAM9 expression for diagnosis. Clinical research incorporating more specimens to assess specificity and sensitivity will also be the focus of future work.

## 5. Conclusions

ADAM9 knockdown suppressed cell proliferation and migration by stopping cells in G0/G1 phase and suppressing EMT. Additionally, from TCGA and GEO study, we found that ADAM9 expression was higher in high-grade cancers. From these results, it can be derived that the expression of ADAM9 is associated with the exacerbation of cancer and correlates with the malignancy.

## Figures and Tables

**Figure 1 biomolecules-12-00791-f001:**
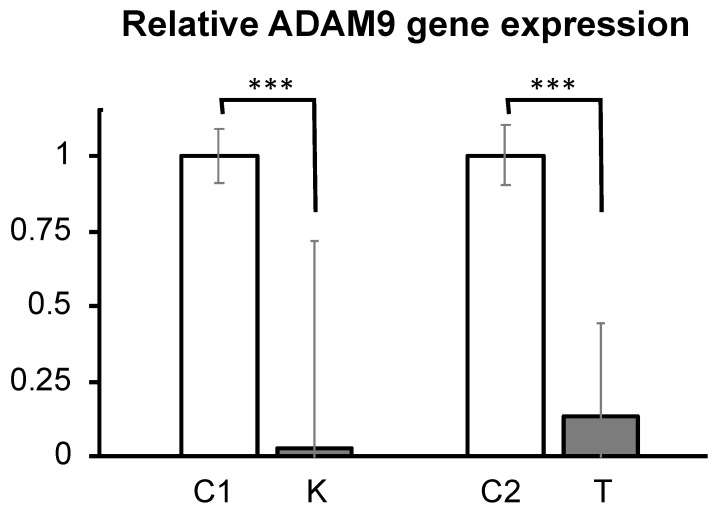
Gene expression of ADAM9 in cell lines. The expressions of ADAM9 were compared in control and ADAM9 knockdown KK47and T24 cells. The expression of ADAM9 significantly decreased in ADAM9 knockdown cell (both KK47 and T24: *** *p* < 0.001) (C1: control of KK47, K: ADAM9 knockdown KK47, C2: control of T24, T: ADAM9 knockdown T24).

**Figure 2 biomolecules-12-00791-f002:**
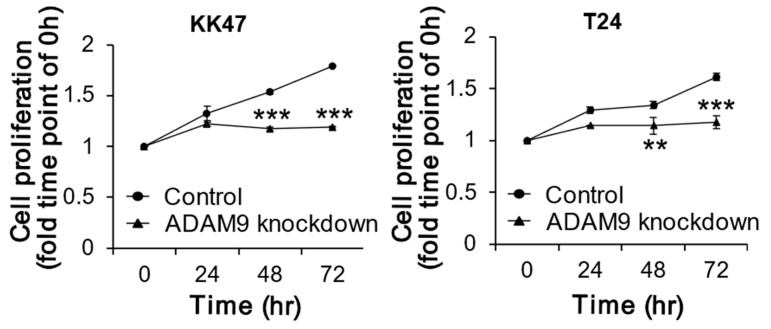
Comparison of cell proliferation between control and ADAM9 knockdown. In vitro cell proliferation assays in KK47 and T24 cell lines treated with Lipofectamine 2000 and 5 nM siRNA against human ADAM9 (n = 3). Group of ADAM9 knockdown significantly inhibited cell proliferation in KK47, and T24 after 48 and 72 h treatment (*p <* 0.05) (** *p* < 0.01, *** *p* < 0.001).

**Figure 3 biomolecules-12-00791-f003:**
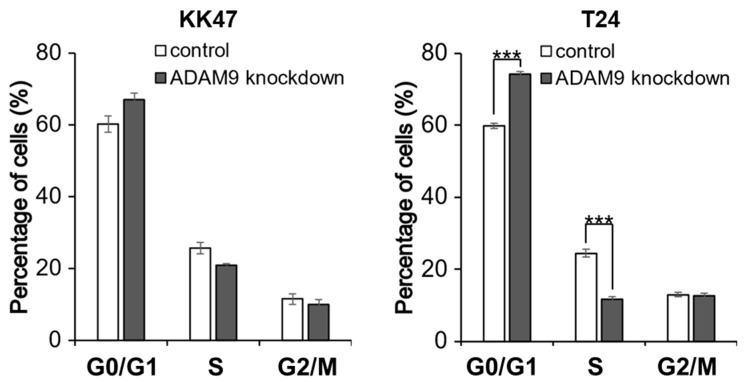
Percentage of cells in each phase. The effects of ADAM9 on cell cycle in KK47 and T24 cells were measured by flow cytometry using Cell Cycle Assay Solution Blue. The percentage of cells in each cycle was investigated in control and ADAM9 knockdown KK47and T24 cell lines. KK47 cells showed no significant difference between control and ADAM9 knockdown, while in T24 cells, G0/G1 phase were increased by ADAM9 knockdown and S phase cells decreased compared with control (*p* < 0.001) (*** *p* < 0.001).

**Figure 4 biomolecules-12-00791-f004:**
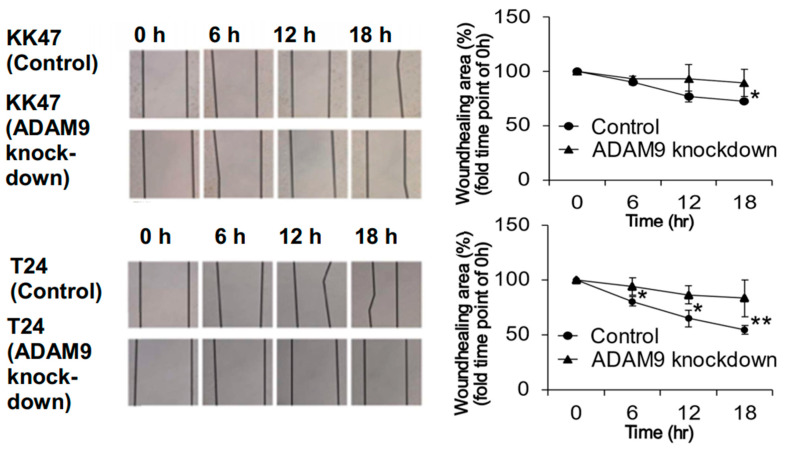
Wound-healing assay in control and ADAM9 knockdown cell lines. Migration ability was investigated in control and ADAM9 knockdown KK47 and T24 cell lines (400×) at 0, 6, 12, and 18 h. Wound area was quantified using ImageJ. ADAM9 knockdown significantly inhibited migration in KK47 cells after 18 h treatment (*p <* 0.05) and in T24 cells after 6, 12, and 18 h treatment (*p <* 0.05) (* *p* < 0.05, ** *p* < 0.01).

**Figure 5 biomolecules-12-00791-f005:**
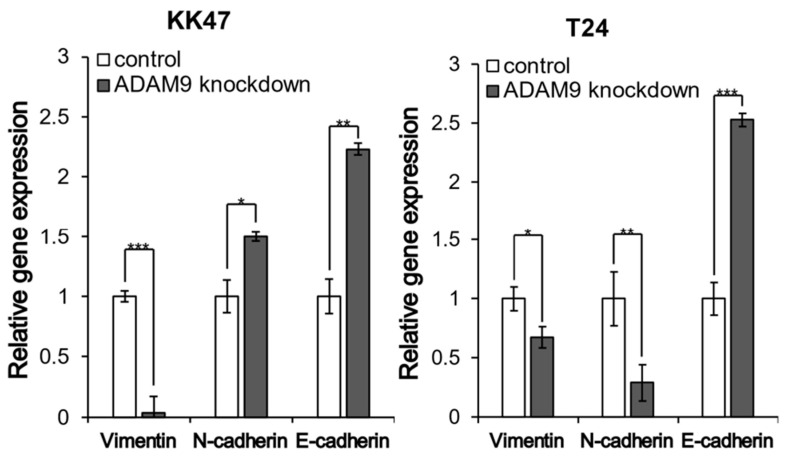
Gene expression of EMT-related proteins. Association between ADAM9 and EMT was investigated by RT-qPCR. The expressions of Vimentin, N-cadherin, and E-cadherin were compared in control and ADAM9 knockdown KK47and T24 cells. In KK47, the expression of vimentin significantly decreased and N-cadherin and E-cadherin significantly increased with ADAM9 knockdown (*p* < 0.05). In T24, the expression of Vimentin and N-cadherin significantly decreased and E-cadherin significantly increased after ADAM9 knockdown (*p* < 0.05) (* *p* < 0.05, ** *p* < 0.01, *** *p* < 0.001).

**Figure 6 biomolecules-12-00791-f006:**
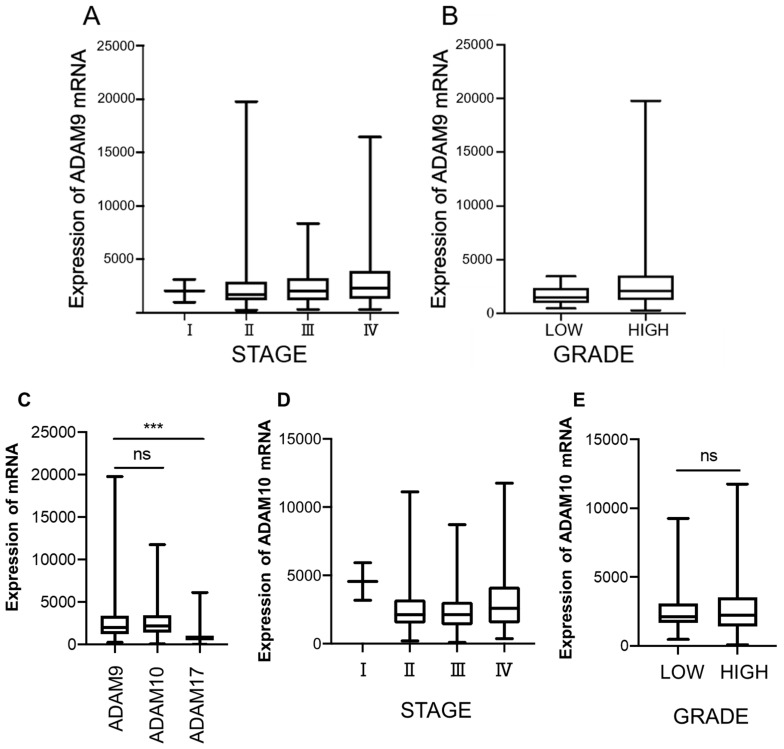
Expression of ADAM9 mRNA in bladder cancer patients: (**A**) the average level of expression of ADAM9 mRNA in stage IV bladder cancer patients was significantly higher than in other stages (*p* < 0.05); (**B**) the expression of ADAM9 mRNA in high-grade bladder cancer was significantly higher than in low-grade cancer (*p* < 0.05); (**C**) the comparison of ADAM9 with ADAM10 and ADAM17 mRNA expression in bladder cancer. The mRNA level of ADAM10 in (**D**) different stages and (**E**) histological grade (*** *p* < 0.0001, ns: no significant difference).

**Figure 7 biomolecules-12-00791-f007:**
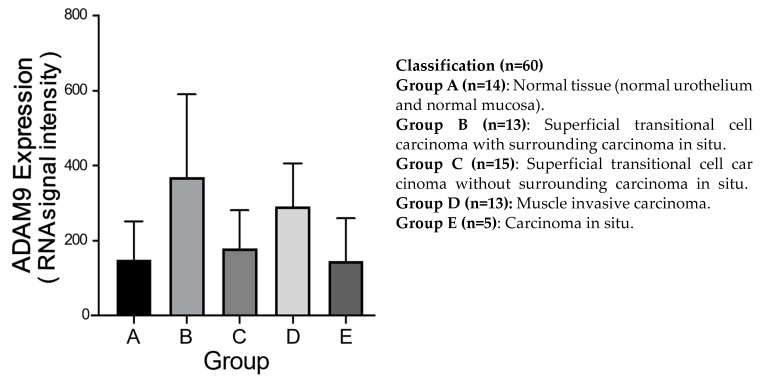
ADAM9 expression in five cancer types. The expression of ADAM9 in Group B (n = 13) was significantly higher than that in Group A (n = 14) and C (n = 15), and that in Group D (n = 13) was significantly higher than those in Group A and E (n = 5) (*p* < 0.001).

**Table 1 biomolecules-12-00791-t001:** Primers used for RT-qPCR.

Gene		Sequences (5′-3′)
ADAM9	Forward	CTTGCTGCGAAGGAAGTACCTG
	Reverse	CACTCACTGGTTTTTCCTCGGC
Vimentin	Forward	GAGAACTTTGCCGTTGAAGC
	Reverse	GCTTCCTGTAGGTGGCAATCT
E-cadherin	Forward	ACGTCGTAATCACCACACTGA
	Reverse	TTCGCTCACTGCTACGTGTAGAA
N-cadherin	Forward	ACAGTGGCCACCTACAAAGG
	Reverse	CCGAGATGGGGTTGATAATG
β-actin	Forward	ATTGCCGACAGGATGCAGAAG
	Reverse	GCTAATCCACATCTGCTGGAA

**Table 2 biomolecules-12-00791-t002:** Row and calculated data using the double delta method (average).

Gene	Cell	CT	ΔCT	ΔΔCT	2^–ΔΔCt^
ADAM9	C1	27.77	−7.733	0	1
	K	32.84	−2.712	5.021	0.03078
	C2	26.27	5.417	0	1
	T	28.98	8.347	2.93	0.1312
Vimentin	C1	23.6	−11.90	0	1
	K	28.4	-7.152	4.751	0.03712
	C2	23.03	2.183	0	1
	T	23.38	2.753	0.57	0.6736
E-cadherin	C1	26.27	−9.227	0	1
	K	25.17	−10.38	−1.158	2.232
	C2	26.63	5.777	0	1
	T	25.07	4.44	1.337	2.526
N-cadherin	C1	30.61	−4.890	0	1
	K	30.08	−5.472	−0.5817	1.497
	C2	26.35	5.5	0	0
	T	27.91	7.283	1.783	0.2905
β-actin	C1	35.5	0		
	K	35.56	0		
	C2	20.85	0		
	T	20.63	0		

(C1: control of KK47, K: ADAM9 knockdown KK47, C2: control of T24, T: ADAM9 knockdown T24).

## Data Availability

The data are available from the corresponding author upon reasonable request.

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
