# Peer review of "Relevance of A Disintegrin and Metalloproteinase Domain-Containing (ADAM)9 Protein Expression to Bladder Cancer Malignancy"

_biomolecules, 2022, doi:10.3390/biom12060791_

Round 1

Reviewer 1 Report

Moriwaki and colleagues performed an interesting study of “Relevance of disintegrin and metalloproteinase domain-containing protein (ADAM) 9 expression to bladder cancer malignancy.” They found that that ADAM9 knockdown suppressed cell proliferation, migration in bladder cancer, and that high-grade bladder cancer is correlated with higher expression of ADAM9. The strength of their study is to elucidate the role of ADAM9 in the progression of bladder cancer and to present its potential as a new therapeutic target.

In this regard, ADAM9 pathway in relation to MHC class I-related chain A (MICA) may provide a strategic ground for cancer immunotherapy. Cell membrane-bound MICA (mMICA) is a ligand for NKG2D, a stimulatory receptor on NK cells. mMICA shedding by ADAM9 or ADAM10 protease is a mechanism that cancer cell escapes the host immune surveillance. The paper was rationally organized with no critical drawbacks, but a few points should be revised.

Minor:
1) The substrates of ADAM proteases such as adhesion molecules and MICA are found be the target of cleavage by ADAM10 and ADAM17. So, I wonder if there are any analysis data related to ADMA10 or ADAM17 other than ADAM9 among your research data of bladder cancer.

2) The role of the ADAM9 knockdown and its relationship with the cancer progression could be better discussed (discuss briefly and cite for instance Kohga K et al, Hepatology 2010(51)1264; or Arai J et al, J Gastroenterol Hepatol 2018(33)1075).

3) In Fig. 7 & Fig. 8, (TCGA data and GEO study) number of samples need to be pointed out.

Author Response

Moriwaki and colleagues performed an interesting study of “Relevance of disintegrin and metalloproteinase domain-containing protein (ADAM) 9 expression to bladder cancer malignancy.” They found that that ADAM9 knockdown suppressed cell proliferation, migration in bladder cancer, and that high-grade bladder cancer is correlated with higher expression of ADAM9. The strength of their study is to elucidate the role of ADAM9 in the progression of bladder cancer and to present its potential as a new therapeutic target.

In this regard, ADAM9 pathway in relation to MHC class I-related chain A (MICA) may provide a strategic ground for cancer immunotherapy. Cell membrane-bound MICA (mMICA) is a ligand for NKG2D, a stimulatory receptor on NK cells. mMICA shedding by ADAM9 or ADAM10 protease is a mechanism that cancer cell escapes the host immune surveillance. The paper was rationally organized with no critical drawbacks, but a few points should be revised.

Minor:
1) The substrates of ADAM proteases such as adhesion molecules and MICA are found be the target of cleavage by ADAM10 and ADAM17. So, I wonder if there are any analysis data related to ADMA10 or ADAM17 other than ADAM9 among your research data of bladder cancer.

(Amendments)

Thank you for the comments.

We have applied the results of ADAM10 and ADAM17 mRNA expression from TCGA dataset (lines 198-202 in page 5-6). We have found that the higher expression of ADAM9 and ADAM10 in bladder cancer compared to ADAM17. Additionally, ADAM10 level was significantly increased in early stage and with no difference between low and high grade of cancer (Fig 6C, D, E in page 9). Hence, we have only focused on ADAM9 in this work. Also, we added to discussion about this at lines 325-333 in page 11.

2) The role of the ADAM9 knockdown and its relationship with the cancer progression could be better discussed (discuss briefly and cite for instance Kohga K et al, Hepatology 2010(51)1264; or Arai J et al, J Gastroenterol Hepatol 2018(33)1075).

(Amendments)

Thank you for the comments.

ADAM proteins have been reported to play an important role in the proteolytic release of transmembrane proteins, including the ectodomain of major histocompatibility complex class I-related chain A (MICA) [26]. ADAM9 knockdown experiments have revealed that ADAM9 is involved in MICA ectodomain shedding of human hepatocellular carcinoma (HCC) cells before [27]. MICA, which is shed from tumor cells, plays a critical role in the immune surveillance against tumor cells and is associated with the prognosis of several malignancies [28]. So, it is suggested that shedding of MICA by ADAM9 may be one of the mechanisms in cell proliferation inhibitory effect and EMT inhibitory effect and the relationship between malignancy and ADAM9 expression obtained in this experiment.

We have added this description at lines 306-315 in page11.

3) In Fig. 7 & Fig. 8, (TCGA data and GEO study) number of samples need to be pointed out.

(Amendments)

Thank you for the comments.

We have added the sample size of TCGA data at line 141 in page 4 and the sample size of GEO study at lines 146 in page 3 and lines 256-257 in page 10 regarding the number of samples and each value.

Reviewer 2 Report

I read the work with great interest, but a few things require clarification:

  1. the methods used in the work are described in very general terms. There is no information on how, for example, the real-time PCR technique was performed. This technique is used several times. In section 2.3 and twice in section 2.7 a reference is made to Table 1 in which the primer sequences are given, but this is not sufficient. No information on the number of repetitions of the reaction, no raw and calculated data using the double delta method.
  2.  Regarding immunochistochemistry, section 2.8 mentions 16 clinical urine cytology from bladder cancer patients. There are no precise data on the histopathological diagnosis. No information on who was included in the control group, no sample/pictures of the results or statistics.
  3. Section 2.11 mentions the comparison of the results with a control group. This group is not described in the paper.

The work is written in a low detail. It is not clear what samples and their number were tested in each analysis. In the description of the results, information about statistical significance appears, but it is not known on what basis it was established, as there are no statistical analyzes. The work cannot be published in this form as it is not possible to verify the conclusions it contains.

Author Response

<Comments and Suggestions for Authors>

  1. The methods used in the work are described in very general terms. There is no information on how, for example, the real-time PCR technique was performed. This technique is used several times. In section 2.3 and twice in section 2.7 a reference is made to Table 1 in which the primer sequences are given, but this is not sufficient. No information on the number of repetitions of the reaction, no raw and calculated data using the double delta method.

(Amendments)

Thank you for the comments.

The PCR conditions were as follows: 94℃ for 2 min followed by 30 cycles of 94°C for 1 min, 61°C for 1 min, and 74°C for 1 min and 30 sec.

We have added this description at lines 121-122 in page 3.

We also have added calculation method at lines 124-137 in page 4.

  1. Regarding immunochistochemistry, section 2.8 mentions 16 clinical urine cytology from bladder cancer patients. There are no precise data on the histopathological diagnosis. No information on who was included in the control group, no sample/pictures of the results or statistics.

(Amendments)

Thank you for the comments.

As the reviewers commented, we have no precise data in clinical urine cytology samples, so we have removed clinical urine cytology examination and evaluation. (original Figure 6, deleted)

  1. Section 2.11 mentions the comparison of the results with a control group. This group is not described in the paper.

(Amendments)

Thank you for the comments.

For the control group, cells that had not been subjected to ADAM9 knockdown treatment were used.

We have added this at lines 93-94 in page 3.

The work is written in a low detail. It is not clear what samples and their number were tested in each analysis. In the description of the results, information about statistical significance appears, but it is not known on what basis it was established, as there are no statistical analyzes. The work cannot be published in this form as it is not possible to verify the conclusions it contains.

(Amendments)

Thank you for the comments.

First, for the TCGA analysis, these description at lines 139-142 in page 4, lines 195-202 in page 5-6 and Fig6 in page 9, and lines 324-333 in page 11. Second, for the GEO study, these description at lines 144-148 in page 4, lines 204-207 in page 6, and Fig7 in page 10, and lines 333-338 in page 11.

Reviewer 3 Report

In this report, Moriwaki et al. evaluated the levels and effects of ADAM9 in bladder cancer. Authors displayed a positive correlation between ADAM9 expression and bladder cancer grade and invasiveness, which is in line with the expression of ADAM9 in other cancers, including prostate, lung and pancreatic. Furthermore, they showed that ablation of ADAM9 expression had effects on migration and EMT of two bladder cancer cell lines (KK47 and T24).

Although the study did not provide deeper insights about the mechanism underlaying the role of ADAM9 in cancer progression, the results are informative and the overall study design is quite sound. Yet, I would require authors to address some points to improve the quality of the manuscript.

Major points:

  1. In my opinion, the introduction section does not provide enough information about ADAM9 and its function in cancer. This should be developed further. In addition, saying that "the purpose of this study was to investigate the role of ADAM9 in bladder cancer..." sounds a bit too ambitious, given the results.
  2. ADAM9 knockdown
    - Line 157, "Binding studies confirmed that ADAM9 was knocked down". Why binding studies? This was proven by a RT-PCR, correct? 
    - Fig 1. Authors should include a control gene (e.g. GAPDH, tubulin, etc) whose expression does not change when cells are treated with  ADAM9 siRNAs.
  3. Immunostaining of ADAM9 in urine cytology samples.
    - In addition to the quantification and statistical analysis, authors should show representative images of the IHC stainings.
    - It is not clear to me, how it is possible that there is no significance in ADAM9 expression between cancer and non-cancer tissues, while the difference is evident between high and low grade cancer.
  4. There must be a problem with the "References", as name of authors are not clear. 
  5. Authors tested the expression of vimentin and cadherins in ADAM9 KD cells and, since downregulation of ADAM9 reduced the expression of vimentin and (partly) promoted the expression of E and N cadherin, they stated that ADAM9 KD suppressed EMT. It is not clear to me if the expression of these 3 genes is an acknowledged manner to investigate EMT. If so, please provide info and related references. Yang et al., in the "Guidelines and definition for research on epithelial-mesenchimal transition" (Nature Reviews Molecular Cell Biology volume 21, pages 341–352 (2020), stated that "EMT status cannot be assessed on the basis of one or a small number of molecular markers". Can authors provide additional information to further prove that ADAM9 affects EMT in bladder cancer cells?

Minor points

  1. Cell cycle analysis. Although a kit was used for this assay, it would be informative for readers briefly including how the assay works.

Author Response

<Comments and Suggestions for Authors>

In this report, Moriwaki et al. evaluated the levels and effects of ADAM9 in bladder cancer. Authors displayed a positive correlation between ADAM9 expression and bladder cancer grade and invasiveness, which is in line with the expression of ADAM9 in other cancers, including prostate, lung and pancreatic. Furthermore, they showed that ablation of ADAM9 expression had effects on migration and EMT of two bladder cancer cell lines (KK47 and T24).

Although the study did not provide deeper insights about the mechanism underlaying the role of ADAM9 in cancer progression, the results are informative and the overall study design is quite sound. Yet, I would require authors to address some points to improve the quality of the manuscript.

Major points:

  1. In my opinion, the introduction section does not provide enough information about ADAM9 and its function in cancer. This should be developed further. In addition, saying that "the purpose of this study was to investigate the role of ADAM9 in bladder cancer..." sounds a bit too ambitious, given the results.

(Amendments)

Thank you for the comments.

ADAM9 has been studied in several cancers. In prostate cancer, ADAM9 cleaves insulin β-chain, tumor necrosis factor (TNF) -α, transforming growth factor (TGF) -α, gelatin, β-casein, and so on, and induces the shedding of epidermal growth factor (EGF), fibroblast growth factor receptor 2IIIB and heparin-binding EGF-like growth factor. Therefore, ADAM9 cause to degrade specific substrates, release active growth factors, interact with key regulatory factors, and its expression is related to several cancer processes including cell growth and invasion [9,10].

We have added this description at lines 65-71 in page 2.

Also, we rewrote as “the purpose of this study was to investigate the involvement of ADAM9 in the exacerbation of bladder cancer” at lines 72-73 in page 2.

  1. ADAM9 knockdown
    - Line 157, "Binding studies confirmed that ADAM9 was knocked down". Why binding studies? This was proven by a RT-PCR, correct? 
    - Fig 1. Authors should include a control gene (e.g. GAPDH, tubulin, etc) whose expression does not change when cells are treated with ADAM9 siRNAs.

(Amendments)

Thank you for the comments.

ADAM9 expression status was confirmed by RT-PCR and we added this at lines 160-162 in pg 5.

And we have changed Fig 1 to a graph exhibiting the results of RT-PCR to quantitatively show that ADAM9 is knocked down. Here, β-actin whose expression levels remain stable in response to treatment with ADAM9 siRNAs was used.

  1. Immunostaining of ADAM9 in urine cytology samples.
    - In addition to the quantification and statistical analysis, authors should show representative images of the IHC stainings:
    - It is not clear to me, how it is possible that there is no significance in ADAM9 expression between cancer and non-cancer tissues, while the difference is evident between high and low grade cancer.

(Amendments)

Thank you for the comments.

As the reviewers commented and mentioned above, we have no precise data in clinical urine cytology samples, so we have removed clinical urine cytology examination and evaluation.

(original Figure 6, deleted).

  1. There must be a problem with the "References", as name of authors are not clear.

(Amendments)

Thank you for the comments.

We have revised the notation of references so that the author’s name can be understood at lines 381-459 in page 13-14.

  1. Authors tested the expression of vimentin and cadherins in ADAM9 KD cells and, since downregulation of ADAM9 reduced the expression of vimentin and (partly) promoted the expression of E and N cadherin, they stated that ADAM9 KD suppressed EMT. It is not clear to me if the expression of these 3 genes is an acknowledged manner to investigate EMT. If so, please provide info and related references. Yang et al., in the "Guidelines and definition for research on epithelial-mesenchimal transition" (Nature Reviews Molecular Cell Biologyvolume 21, pages 341–352 (2020), stated that "EMT status cannot be assessed on the basis of one or a small number of molecular markers". Can authors provide additional information to further prove that ADAM9 affects EMT in bladder cancer cells?

(Amendments)

Thank you for the comments.

We agree with the comments, so we have added the study limitation as follows (lines 341-343 in page 12).

Our story of ADAM9 knockdown and EMT may be in part conclusive because Yang et al. stated that EMT status cannot be assessed on the basis of one or a small number of molecular marker [31].

EMT is related to bladder cancer metastatic and EMT is characterized by upregulation of Vimentin and N-cadherin and downregulation of E-cadherin [21, 22].

Minor points

Cell cycle analysis. Although a kit was used for this assay, it would be informative for readers briefly including how the assay works.

(Amendments)

Thank you for the comments.

We added a more detailed description of how to conduct cell cycle assay at lines 102-108 in page 3.

The added contents are as follows.

The effects of ADAM9 on the cell cycle in KK47 and T24 cells were measured using Cell Cycle Assay Solution Blue (DOJINDO, Japan). After ADAM9 knockdown, 1ml cell suspension having 1×105 cells/ml was prepared (n = 3) and centrifuged at 300g for 5 min. The supernatant was removed, 1 ml PBS was added, and after resuspension, was centrifuged at 300 g for 5 min. The supernatant was removed again, then 0.5 ml PBS and 5 μl cell cycle assay solution were added. After incubation for 15 min at 37℃ in shade cells in each cell cycle were measured by flow cytometry.

References

[9]  Kim JM, Jeung HC, Rha SY, Yu EJ, Kim TS, Shin YK, Zhang X, Park KH, Park SW, Chung HC, Powis G. The effect of disintegrin-metalloproteinase ADAM9 in gastric cancer progression. Mol Cancer Ther. 2014;13(12):3074-85. doi: 10.1158/1535-7163.MCT-13-1001.

[10] Oria VO, Lopatta P, Schmitz T, Preca BT, Nyström A, Conrad C, Bartsch JW, Kulemann B, Hoeppner J, Maurer J, Bronsert P, Schilling O. ADAM9 contributes to vascular invasion in pancreatic ductal adenocarcinoma. Mol Oncol. 2019;13(2):456-479. doi: 10.1002/1878-0261.12426.

[21] Loh CY, Chai JY, Tang TF, Wong WF, Sethi G, Shanmugam MK, Chong PP, Looi CY. The E-Cadherin and N-Cadherin Switch in Epithelial-to-Mesenchymal Transition: Signaling, Therapeutic Implications, and Challenges. Cells. 2019;8(10):1118. doi: 10.3390/cells8101118.

[22] Usman S, Waseem NH, Nguyen TKN, Mohsin S, Jamal A, Teh MT, Waseem A. Vimentin Is at the Heart of Epithelial Mesenchymal Transition (EMT) Mediated Metastasis. Cancers (Basel). 2021;13(19):4985. doi: 10.3390/cancers13194985.

[26] Arai J, Goto K, Stephanou A, Tanoue Y, Ito S, Muroyama R, Matsubara Y, Nakagawa R, Morimoto S, Kaise Y, Lim LA, Yo-shida H, Kato N. Predominance of regorafenib over sorafenib: Restoration of membrane-bound MICA in hepatocellular carcinoma cells. J Gastroenterol Hepatol. 2018;33(5):1075-1081. doi: 10.1111/jgh.14029.

[27] Kohga K, Takehara T, Tatsumi T, Ishida H, Miyagi T, Hosui A, Hayashi N. Sorafenib inhibits the

shedding of major histo-compatibility complex class I-related chain A on hepatocellular carcinoma cells by down-regulating a disintegrin and met-alloproteinase 9. Hepatology. 2010;51(4):1264-73. doi: 10.1002/hep.23456.

[28] Zhang X, Yan L, Jiao W, Ren J, Xing N, Zhang Y, Zang Y, Wang J, Xu Z. The clinical and

biological significance of MICA in clear cell renal cell carcinoma patients. Tumour Biol.

2016;37(2):2153-9. doi: 10.1007/s13277-015-4041-7.

[31] Yang J, Antin P, Berx G, Blanpain C, Brabletz T, Bronner M, Campbell K, Cano A, Casanova J, Christofori G, Dedhar S, Derynck R, Ford HL, Fuxe J, García de Herreros A, Goodall GJ, Hadjantonakis AK, Huang RYJ, Kalcheim C, Kalluri R, Kang Y, Khew-Goodall Y, Levine H, Liu J, Longmore GD, Mani SA, Massagué J, Mayor R, McClay D, Mostov KE, Newgreen DF, Nieto MA, Puisieux A, Runyan R, Savagner P, Stanger B, Stemmler MP, Takahashi Y, Takeichi M, Theveneau E, Thiery JP, Thompson EW, Weinberg RA, Williams ED, Xing J, Zhou BP, Sheng G; EMT International Association (TEMTIA). Guidelines and definitions for research on epithelial-mesenchymal transition. Nat Rev Mol Cell Biol. 2020;21(6):341-352. doi: 10.1038/s41580-020-0237-9.

Round 2

Reviewer 2 Report

not clear how the tests were done and what were used as controls

Author Response

We did ADAM9 knowdown, then saw the effect on 1. cell proliferation,
2. cell cycle in order to see which cell cycle step ADAM9 plays a role, 3. wound healing assay in order to see how ADAM9 affects on cell invasion or EMT, 4. EMT marker expressions by RT-PCR (4 kinds of analysis).

Then we concluded that ADAM9 knockdown demonstrated the inhibition
of noninvasive and invasive bladder cancer cell lines proliferation change cell cycle (G0/G1 phase increase and S phase decrease) in
invasive bladder cancer cells (T24), inhibition of cell invasion
especially in invasive bladder cancer cells (T24). and blocked EMT
especially in T24 in RT-PCR.

Next, using data-base such as  TCGA analysis and GEO study in
bladder cancer cell lines, ADAM9 mRNA is expressed in higher stage
and higher grade bladder cancer samples.

Taken together (in vitro experiments and in vivo ones using
data-base setting), ADAM9 relates to cell proliferation affecting
cell cycle and EMT in especially invasive bladder cancer cells and
were verified in clinical bladder cell samples (high stage and high
grade) using data-base setting study.

Therefore, ADAM9 may play a role in invasion or progression in
invasive bladder cancer.

The raw data from real time PCR is added (page5-page6). 

Reviewer 3 Report

The authors have amended the manuscript as suggested.

Author Response

thanks